# Seeking Medical Assistance for Dysphonia Is Associated with an Improved Survival Rate in Laryngeal Cancer: Real-World Evidence

**DOI:** 10.3390/diagnostics11020255

**Published:** 2021-02-07

**Authors:** Yi-An Lu, Ming-Shao Tsai, Li-Ang Lee, Shu-Ru Lee, Li-Yun Lin, Chain-Fen Chang, Wan-Ni Lin, Li-Jen Hsin, Chun-Ta Liao, Hsueh-Yu Li, Yu-Wen Wen, Tuan-Jen Fang

**Affiliations:** 1Department of Otolaryngology Head and Neck Surgery, Linkou Chang Gung Memorial Hospital, No. 5 Fushing St., Taoyuan 333, Taiwan; b9402009@cgmh.org.tw (Y.-A.L.); 5738@cgmh.org.tw (L.-A.L.); liyun915@cgmh.org.tw (L.-Y.L.); 409446095@m365.fju.edu.tw (C.-F.C.); y1829@cgmh.org.tw (W.-N.L.); d7239@cgmh.org.tw (L.-J.H.); liaoct@cgmh.org.tw (C.-T.L.); hyli38@cgmh.org.tw (H.-Y.L.); 2College of Medicine, Chang Gung University, Taoyuan 333, Taiwan; b87401061@cgmh.org.tw; 3Department of Otolaryngology Head and Neck Surgery, Chiayi Chang Gung Memorial Hospital, Chiayi 613, Taiwan; 4Research Services Center for Health Information, Chang Gung University, Taoyuan 333, Taiwan; s19880711@mail.cgu.edu.tw; 5Clinical Informatics and Medical Statistics Research Center, Chang Gung University, Taoyuan 333, Taiwan; ywwen@mail.cgu.edu.tw

**Keywords:** laryngeal cancer, glottic cancer, dysphonia, hoarseness, overall survival

## Abstract

(1) Background: Patients with laryngeal cancer usually present with dysphonia. However, some studies reported that the duration from dysphonia to cancer diagnosis has been prolonged significantly in recent years. This study aimed to evaluate that in the initial dysphonia-related diagnosis and the interval between the diagnosis of laryngeal cancer may affect the overall survival (OS). (2) Methods: The 1997–2013 Longitudinal Health Insurance Database was used in this study. A propensity score with 1-to-1 matching was applied to balance the baseline characteristics. The OS was examined by the Kaplan-Meier method and log-rank test. (3) Results: A total of 2753 patients with a first primary laryngeal cancer diagnosis were identified. The patients without prior dysphonia-related diagnosis (PD−) group did have a significantly worse five-year survival (*p* = 0.015) comparing with those with a prior dysphonia-related diagnosis (PD+) group among glottic cancer patients. The group with a shorter dysphonia-to-diagnosis interval had a better five-year OS than the prolonged group (*p* = 0.007) in laryngeal cancer. (4) Conclusions: Looking for medical assistance before a diagnosis of glottic cancer is associated with a better overall survival, while a diagnostic delay of more than 30 days from the first medical examination for dysphonia is associated with a worse outcome among in patients with laryngeal cancer.

## 1. Introduction

Laryngeal cancer, accounting for 21.6% of all new head and neck cancer cases in Western countries, is the second-most common head and neck malignancy [1]. Patients with laryngeal cancers may present with persistent hoarseness early in their disease. Vocal fold vibration or resonance is influenced by the growth of laryngeal cancer tumors, which results in a voice change. However, the hoarseness may be ignored because the causes of impaired voice vary from inflammation to malignancies. There is still a lack of evidence to suggest when a patient with hoarseness should be referred for a thorough larynx examination.

Preserving laryngeal function is an important goal in treating laryngeal cancer. It is hard to preserve a patient’s voice and swallowing functions in cases of advanced laryngeal cancers, regardless of the treatment modalities. Advanced stage laryngeal cancer had a poor prognosis, with five-year overall survival (OS) rate of 36–55% while more than 80% survive longer than five-years in early-stage cases [2,3,4]. Delays in both access to care and diagnosis have been associated with a worse prognosis in laryngeal cancer [5].

In the United States, the delayed presentation of patients with laryngeal cancer to primary care physicians and otolaryngologists has significantly increased in recent years [6]. The duration from dysphonia to cancer diagnosis has been prolonged significantly. However, the impact of a delayed diagnosis on outcomes has not been reported. In the present study, we aim to evaluate the impact of preexisting dysphonia and its duration on the outcomes of laryngeal cancers from our national cancer registration database. We hypothesize that delay in making the diagnose of laryngeal cancer from the initial dysphonia-related diagnosis may decrease OS.

## 2. Materials and Methods

This study was conducted by using the National Insurance Research Database (NHIRD), which was established in 1995 and covered more than 99% of the whole population in Taiwan [7]. Claims for inpatient, outpatient and emergency visits, procedures, hospitalizations and prescription medications were recorded in the NHIRD. For each visit, the date, prescriptions and diagnosis codes based on the International Classification of Disease, 9th Edition (ICD-9 CM codes) were recorded in the database which have been confirmed to be of high quality [8,9]. We used the 1997–2013 Longitudinal Health Insurance Database, including one million individuals from Taiwan’s population who were randomly drawn from the NHIRD. All aspects of the study were approved by the Human Studies Research Committee of Chang Gung Medical Foundation. (201506227B0, 14 October 2015).

From the data recorded between 1 January 1997 and 31 December 2012, the patients with a first primary laryngeal cancer were identified by the following ICD-9 CM codes: 161.0, 161.1, 161.2, 161.3, 161.8, 161.9). Among the patients with a first primary laryngeal cancer, the patients with outpatient or inpatient visit for hoarseness within six months before diagnosis of laryngeal cancer (ICD-9 CM codes: 212.1, 352.3, 438.10, 438.19, 464, 464.01, 464.20, 464.21, 476.0, 476.1, 478.30, 478.31, 478.32, 478.33, 478.34, 478.4, 478.5, 478.6, 478.70, 478.71, 478.75, 478.79, 784.40, 784.41, 784.42, 784.49) were identified. The patients were classified into the prior dysphonia-related diagnosis (PD+) group, and those without a prior dysphonia-related diagnosis (PD−) within six months before their ultimate diagnosis of laryngeal cancer were classified into the PD− group. (Figure 1) It is worth noting that the dysphonia-related diagnosis within this database may not relate to laryngeal cancer. For example, if someone in the database had uncomplicated laryngitis in 1999 and then had an entirely unrelated laryngeal cancer diagnosis in 2009, then this patient would end up in the PD+ PL group. Therefore, we defined prior dysphonia-related diagnosis within six months as presumable related-cancer-dysphonia.

A delay in diagnosis was defined as greater than 30 days from identification of a lesion to the diagnostic test in previous literature [10]. We further classified PD+ group into two groups: the immediate (IM) and prolonged (PL) groups. The IM group was defined by a dysphonia-to-diagnosis (of a first primary laryngeal cancer) interval (DDI) equal to or less than 30 days, while the PL group was defined as a DDI over 30 days. The baseline characteristics, including sex, age, and health condition variables (including chronic cardiovascular diseases, chronic cerebrovascular diseases, chronic lung diseases, chronic renal diseases, diabetes mellitus, liver cirrhosis, rheumatologic diseases, malignancy, HIV infection, organ transplantation, and Charlson Comorbidity Index (CCI)) were also collected.

The primary outcome was OS. Each of the patients was followed from the date of diagnosis with a first primary laryngeal cancer to the date of death or the end of the database (31 December 2013); all of the patients were followed for at least one year.

### Statistical Analysis

Baseline characteristics are presented as the mean ± standard deviation or a percentage. Continuous variables were analyzed by Student’s *t*-test, and categorical variables were analyzed by the chi-squared test. The Wilcoxon rank sum test and Wilcoxon signed-rank test were applied to independent continuous data and paired continuous data, respectively if the assumption of normal distribution is violated. To reduce the selection bias due to an imbalance of baseline characteristics between the comparison groups, a propensity score method with 1-to-1 matching was applied. The goal of propensity scores matching is to approximate a random experiment. The propensity scores for all patients were obtained by implementing a multiple logistic regression. All variables in the propensity score model are shown in Table 1. After propensity score matching, the balance of the characteristics was examined by standardized mean difference and variance. The OS was examined by the Kaplan-Meier method and the log-rank test. Subgroup analysis for the patients with glottic cancer (ICD-9 CM Code: 161.0) was also performed. All analyses were performed using SAS software, version 9.4 (SAS Institute Inc., Cary, NC, USA). The statistical significance was defined as a two-sided *p*-value < 0.05.

## 3. Results

### Subsection

A total of 2753 patients with first primary laryngeal cancer were identified and included in this study. There were 944 patients diagnosed with dysphonia before their cancer diagnosis, and 1809 patients did not. Among the 944 patients with dysphonia, approximately 53% (*n* = 499) and 47% (*n* = 445) of their DDIs were ≤30 days and >30 days, respectively. The sample selection is shown in Figure 1.

From 1999 to 2013, 2753 patients were diagnosed with laryngeal cancer; 479 patients were female (17.4%), and 2274 were male (82.6%) (Table 1). Compared with the PD+ group, patients in the PD− group were younger (56.91 ± 15.11 vs. 61.15 ± 12.66; *p* < 0.001) but had a worse general health status (CCI 1.75 ± 2.41 vs. 1.38 ± 2.02; *p* < 0.001). Patients in the PD− group had less diabetes (*p* < 0.001) and chronic cardiovascular diseases (*p* < 0.001) but more liver cirrhosis (*p* = 0.016) and second primary malignancies (*p* < 0.001). We further divided the patients according to their cancer subsites. When focusing on the glottic cancer patients (Table 2), the patients in the PD− group still had a worse general health status (CCI 2.71 ± 2.78 vs. 1.70 ± 1.84; *p* < 0.001). The PD− group included more patients with chronic lung diseases (*p* = 0.016) and second primary malignancies (*p* < 0.001).

After propensity score matching, we found that the PD+ group had significantly lower risk of mortality than the PD− group (HR = 0.78 (0.64–0.96)). In the IM group, the risk of mortality was significantly lower than that in the PL group (HR = 0.70 (95% CI: 0.51–0.95)) (Table 3)

When comparing the survival differences between the PD+ group and the PD− group, the PD− group had a decreasing trend in the five-year OS compared to the PD+ group, but there was no significant difference (*p* = 0.117, 77% and 74% in five-years survival) (Figure 2A). For the patients whose tumors originated from the glottic area, the PD− group showed significantly worse five-year OS (*p* = 0.015, 78% and 68% in five-years survival) (Figure 2B). In the investigation of the impact of DDI on five-year OS rates in laryngeal cancer patients, we found that the IM group had a better five-year OS than the PL group (*p* = 0.007, 81% and 73% in five-years survival) (Figure 2C), but there were no differences in the glottic subgroup (*p* = 0.76, 79% and 76% in five-years survival) (Figure 2D).

## 4. Discussion

Laryngeal cancer is a common head and neck cancer, with an estimated 210,000 new cases diagnosed yearly worldwide [11]. The laryngeal cancer prognostic factors that impact survival include patient factors such as age, sex, and comorbidities as well as tumor factors such as clinical tumor characteristics and nodular stage [12]. Recently, systemic factors including delayed referral, delayed treatment and insurance status have been studied but not yet associated with survival [13]. Hoarseness was the most common symptom regardless the tumor subsite of laryngeal cancer. In glottic cancer, the first symptom in over 95% of patients is vocal change [14]. These voice symptoms make laryngeal malignancies possible to be detected in the early stage. Thus, this study suggested laryngeal cancer patients with previous dysphonia-related diagnosis had better outcomes.

Patients with dysphonia before the diagnosis of glottic cancer showed better OS in our study. Among patients with glottic tumors, dysphonia usually occurs while the lesions are small [15]. Although the tumor stage was noted recorded in the NHIRD, we proposed that patients can be warned by such impairment at an early stage and thus improved the outcomes. However, there was no significant difference in the laryngeal cancer group. In cases of laryngeal cancers that originate in the supra-or subglottis, the voice change happens when the tumor invades the glottic area [14]. Thus, dysphonia in supra- or subglottic cancer does not warn patients when tumor is small unlike the voice symptoms leading to the early detection of patients with glottic cancer [16]. The dysphonia symptoms influence patients’ outcomes more in glottic cancer than other subsites of laryngeal cancer [17].

Patients without a dysphonia-related diagnosis in our study were younger but had a worse general health status and a higher incidence of second primary malignancy. Smoking is a well-known risk factor for the development of malignancies in the head and neck region [18]. A Japanese study reported that the incidence of smoking was higher among patients with glottic cancer without dysphonia than among patients with voice complaints [19]. Smokers have chronic voice changes related to vocal edema or reflux laryngitis. A heavy smoker may be less sensitive to small vocal lesions than nonsmokers. Thus, smokers may be more prone to be diagnosed with glottic cancer without previous dysphonia. In our study, NHIRD had the inherent limitations of missing demographic information, such as smoking and vocal demand. We did not assume that being “without hoarseness” meant that patients had a normal voice quality. The voice demand is related to individual factors, such as sex, age, profession and tobacco use [20]. Patients with chronic vocal abuse or with a smoking habit may be accustomed to their impaired voice and delay their search for medical help [21]. Thus, the diagnosis of dysphonia reflects not only the voice quality itself but the level of concern self-wellness. Our results showed more second primaries in the PD− group, which may also imply that they may have become accustomed to their voice after disease treatment or because of more other health problems.

Tachibana et al. reported that glottic cancer patients without hoarseness had a better disease-specific survival than those with hoarseness [19]. They claimed that glottic cancer patients without complaints of hoarseness were diagnosed at earlier stages. In this single-center study, the glottic cancer cases were identified by the department of otolaryngology; 78% of the cases were identified by fiberscopes and 21.9% of the cases was found during gastroscopic or bronchoscopic examinations. The high proportion of patients without dysphonia in this cohort may be due to the distribution of the patients’ origins. The incidental findings of laryngeal lesions during endoscopy screening increase the rate of diagnoses at the earliest stages in the patient population. In contrast, the present study is based on nationwide and papulation-based real-world data, including cases from all over the country. The distribution of patient characteristics and presentations are more representative. But we concern regular health examination, appropriate referrals to specialists and close follow-up by otolaryngologists may increase the early discovery of malignancy in patients without symptoms of hoarseness [22].

Among our laryngeal cancer patients with dysphonia, patients with a DDI longer than 30 days had a worse five-year OS. Previous studies revealed that delayed diagnosis not only resulted in tumor progression but also increased healthcare costs [23]. In cases of head and neck squamous cell carcinoma, a 4-week delay in treatment resulted in a significant increase in the risk of local failure, neck failure and decreased survival [23]. Cohen et al. reported that, compared to patients who saw an otolaryngologist more than 1 month after a referral from their general physician, those who complained of laryngeal symptoms and were referred within 1 month had lower healthcare costs [24]. The possibility of laryngeal malignancy must be kept in mind when patients have prolonged hoarseness. According to our study, early referral and laryngoscopy in high risk patients was suggested.

Since the development of the laryngeal fiberscope, the diagnosis of stage I to II disease among all patients with glottic cancer has increased significantly [25]. It was also found that the patients diagnosed with laryngeal cancer after follow-up for chronic laryngitis by an ENT specialist had a better overall prognosis, mostly because of the more favorable stage at presentation [5]. Laryngoscopy or a referral for laryngoscopy to an otolaryngologist should be a considered for patients with persistent dysphonia due to the low risk and high benefit of the procedure itself. For suspicious lesions included leukoplakia, erythroplakia, ulceration, exophytic mass, and a lesion on an immobile vocal fold, follow-up with a fiberscope and laryngeal biopsy may be done in high-risk patients who smoke, drink alcohol or have a family history of cancer.

This study had some limitations. First, are the inherent limitations of NHIRD data quality, including the lack of tumor histology and staging. The NHIRD also does not offer demographic information, such as smoking, vocal demand, and medical factors that may influence patient outcomes. Second, we defined PD+ group as prior dysphonia-related diagnosis within six months before their ultimate diagnosis of laryngeal cancer to reduce the bias. However, the limitation of difficult to distinguish potentially unrelated-cancer-dysphonia was still existed. Third, the physician may not have coded for dysphonia if the patient had laryngeal cancer, so dysphonia may be significantly under-reported. Nevertheless, NHIRD is a nationally established database that covers 99.6% of Taiwan’s population [26]; the large sample size lends adequate statistical power to evaluate differences. The results reflect the real-world status in general population and can be widely applied clinically.

## 5. Conclusions

Looking for medical assistance before a diagnosis of glottic cancer is associated with a better overall survival, while a diagnostic delay of more than 30 days from the first medical examination for dysphonia is associated with a worse outcome among in patients with laryngeal cancer. Early referral to an otolaryngologist for laryngoscopy should be a “strong recommendation” in high risk patients with persistent voice symptoms.

## Figures and Tables

**Figure 1 diagnostics-11-00255-f001:**
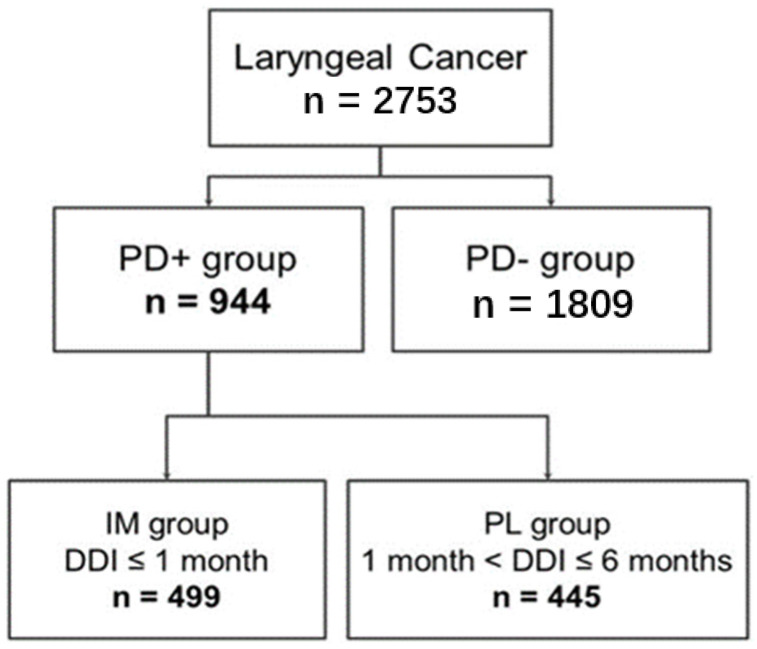
Study patient enrollment flowchart of this study. Abbreviations: DDI: dysphonia-to-diagnosis (of a first primary laryngeal cancer) interval; IM group: immediate group; PD+ group: prior dysphonia-related diagnosis group; PD−group: without a prior dysphonia-related diagnosis group; PL group: prolonged group.

**Figure 2 diagnostics-11-00255-f002:**
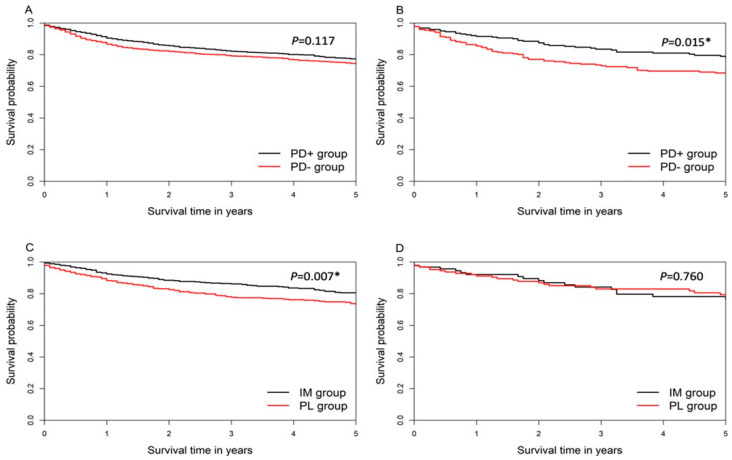
(**A**) five-year overall survival between PD+ group and PD− group in laryngeal cancer patients (**B**) five-year overall survival PD+ group and PD− group in glottic cancer patients (**C**) five-year overall survival between IM group and PL group in laryngeal cancer (**D**) five-year overall survival between IM group and PL group in glottic cancer patients. *: *p* < 0.05; Abbreviations: IM group: immediate group; PD+ group: prior dysphonia-related diagnosis group; PD− group: without a prior dysphonia-related diagnosis group; PL group: prolonged group.

**Table 1 diagnostics-11-00255-t001:** Laryngeal cancer patient demographics and comorbidity.

	Laryngeal Cancer						
	Total Sample		PD− Group		PD+ Group		*p*-Value
	(*n* = 2753)		(*n* = 1809)		(*n* = 944)		
Age (mean ± SD)	58.37 ± 14.45		56.91 ± 15.11		61.15 ± 12.66		<0.001 *
Sex							<0.001 *
Female, (n%)	479	(17.40%)	386	(21.34%)	93	(9.85%)	
Male, (n%)	2274	(82.60%)	1423	(78.66%)	851	(90.15%)	
CCI (mean ± SD)	1.63 ± 2.29		1.75 ± 2.41		1.38 ± 2.02		<0.001 *
Comorbidity, (n%)							
Chronic cardiovascular diseases	603	(21.90%)	346	(19.13%)	257	(27.22%)	<0.001 *
Chronic cerebrovascular diseases	217	(7.88%)	134	(7.41%)	83	(8.79%)	0.2006
Chronic lung diseases	522	(18.96%)	325	(17.97%)	197	(20.87%)	0.0652
Chronic renal diseases	115	(4.18%)	78	(4.31%)	37	(3.92%)	0.6255
Diabetes mellitus	367	(13.33%)	211	(11.66%)	156	(16.53%)	<0.001 *
Liver cirrhosis	93	(3.38%)	72	(3.98%)	21	(2.22%)	0.0155 *
Rheumatologic diseases	56	(2.03%)	37	(2.05%)	19	(2.01%)	0.9541
Malignancy	842	(30.58%)	658	(36.37%)	184	(19.49%)	<0.001 *

*: *p* < 0.05; Abbreviations: CCI: Charlson Comorbidity Index; PD+ group: prior dysphonia-related diagnosis group; PD− group: without a prior dysphonia-related diagnosis group.

**Table 2 diagnostics-11-00255-t002:** Glottic cancer patient demographics and comorbidity.

	Glottic Cancer						
	Total Sample		PD− Group		PD+ Group		*p*-Value
	(*n* = 755)		(*n* = 295)		(*n* = 460)		
Age (mean ± SD)	63.33 ± 12.85		62.75 ± 14.26		63.70 ± 11.86		0.5450
Sex							<0.001 *
Female, (n%)	78	(10.33%)	45	(15.25%)	33	(7.17%)	
Male, (n%)	677	(89.67%)	250	(84.75%)	427	(92.83%)	
CCI (mean ± SD)	2.10 ± 2.31		2.71 ± 2.78		1.70 ± 1.84		<0.001 *
Comorbidity, (n%)							
Chronic cardiovascular diseases	201	(26.62%)	72	(24.41%)	129	(28.04%)	0.2706
Chronic cerebrovascular diseases	72	(9.54%)	28	(9.49%)	44	(9.57%)	0.9732
Chronic lung diseases	144	(19.07%)	69	(23.39%)	75	(16.30%)	0.0156 *
Chronic renal diseases	24	(3.18%)	10	(3.39%)	14	(3.04%)	0.7916
Diabetes mellitus	111	(14.70%)	35	(11.86%)	76	(16.52%)	0.08
Liver cirrhosis	23	(3.05%)	13	(4.41%)	10	(2.17%)	0.0817
Rheumatologic diseases	13	(1.72%)	6	(2.03%)	7	(1.52%)	0.6
Malignancy	161	(21.32%)	105	(35.59%)	56	(12.17%)	<0.001 *

*: *p* < 0.05; Abbreviations: CCI: Charlson Comorbidity Index; PD+ group: prior dysphonia-related diagnosis group; PD− group: without a prior dysphonia-related diagnosis group.

**Table 3 diagnostics-11-00255-t003:** Hazard Ratio between PD+/PD− and IM/PL groups in laryngeal cancer and glottic cancer after propensity score matching.

	Hazard Ratio (95% CI)	*p*-Value
Laryngeal cancer		
PD+ group	0.78 (0.64–0.96)	0.0172 *
PD− group	ref	
PD+ group		
IM group	0.70 (0.51–0.95)	0.0233 *
PL group	ref	
PD+ group	0.54 (0.37–0.82)	0.0037 *
PD− group	ref	
PD+ group		
IM group	1.17 (0.60–2.29)	0.6458
PL group	ref	

* *p* < 0.05; Abbreviations: IM group: immediate group; PD+ group: prior dysphonia-related diagnosis group; PD− group: without a prior dysphonia-related diagnosis group; PL group: prolonged group.

## Data Availability

The datasets generated or analyzed in the current study can be accessed from the Taiwan National Health Insurance Research Database repository (https://nhird.nhri.org.tw/en/How_to_cite_us.html).

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
