# Peer review of "Seeking Medical Assistance for Dysphonia Is Associated with an Improved Survival Rate in Laryngeal Cancer: Real-World Evidence"

_diagnostics, 2021, doi:10.3390/diagnostics11020255_

Round 1
Reviewer 1 Report
The authors included all dysphonia diagnoses (benign and non-benign) for comparison. Although this is a good first step, a more valuable analysis would be subgrouping benign vs. non-benign diagnoses within the long (5 years) and narrow (6 months) time frame against survival rates among patients with laryngeal cancer. The subgroup analysis appears to have completed this comparison as well as more in-depth overall health status.
Number of subjects in this study is impressive, and valuable as a national finding. The finding of worse health status among younger subjects diagnosed with laryngeal cancer is quite interesting. Further analysis (in subsequent study) should examine environmental risk factors (urban vs. rural domicile, smoker vs. nonsmoker). Hopefully, the NHIRD has this data available for further analysis.
The very real need to be evaluated within 4 weeks of dysphonia symptoms is important to note as countries make health care more, or less, available to their citizens. This should be part of ever nation’s health care policy.
Author Response
Dear esteemed editor and reviewers,
We are very grateful of the comments given by the editor and reviewers. The point-to-point correspondence to each comment is listed as follows. We hope the revised manuscript is suitable for publication in Diagnostics.
Sincerely yours,
Tuan-Jen Fang, MD
Chang Gung Memorial Hospital, No. 5 Fushing St., Taoyuan 333, Taiwan
Tel.: +(886)-3328-1200 ext. 3846; Fax: +(886)-3328-1200 ext. 2667;
E-mail: [email protected]
Response to Reviewer 1 Comments
Point 1: The authors included all dysphonia diagnoses (benign and non-benign) for comparison. Although this is a good first step, a more valuable analysis would be subgrouping benign vs. non-benign diagnoses within the long (5 years) and narrow (6 months) time frame against survival rates among patients with laryngeal cancer. The subgroup analysis appears to have completed this comparison as well as more in-depth overall health status.
Response 1: Thank you for bringing this to us. Our study focuses on seeking medical assistance within 6 months before cancer diagnosis and resulted significant finding. If we investigate longer (5 years) time frame before cancer diagnosis, the inclusion and exclusion criteria have to be modified. The patients who were diagnosed during 1997-2002 should be excluded due to the lack of 5 years backtracking before cancer diagnosis. We may investigate this issue in further research. Thank you for the impressive suggestion for evaluation of more in-depth overall health status.
Point 2: Number of subjects in this study is impressive, and valuable as a national finding. The finding of worse health status among younger subjects diagnosed with laryngeal cancer is quite interesting. Further analysis (in subsequent study) should examine environmental risk factors (urban vs. rural domicile, smoker vs. nonsmoker). Hopefully, the NHIRD has this data available for further analysis.
Response 2: Thank you for reminding us. The information of smoking is not available in NHIRD. We descripted this limitation in discussion as below.
“The NHIRD also does not offer demographic information, such as smoking, vocal demand, and medical factors that may influence patient outcomes” (line 264-266)
The analysis of urban vs. rural domicile is still NHIRD’s limitation. The place of insurance was recorded in NHIRD which may not equal to place of residence in case the patient insured following his/her family dependents. It’s interesting to investigate the subgroup between urban and rural domicile. We are looking forward further result will be investigated in other databases.
The very real need to be evaluated within 4 weeks of dysphonia symptoms is important to note as countries make health care more, or less, available to their citizens. This should be part of ever nation’s health care policy.
Reviewer 2 Report
Thank you for the opportunity to review “Seeking Medical Assistance for Dysphonia is Associated with an Improved Survival Rate in Laryngeal Cancer: Real-world Evidence”
In this work the authors analyzed retrospectively, basing on a large database, a cohort of patients with a diagnosis of first primary laryngeal cancer. The authors conclude that looking for medical assistance for dysphonia improves OS, on the contrary a lag of more than 30 days between the first dysphonia-related diagnosis and laryngeal cancer diagnosis is associated with worse OS rates.
The work is overall fairly written. Anyway, English should be revised and corrected, since some sentences are quite confusing and not correctly built. Results seem interesting and confirm the importance of an early diagnosis for laryngeal cancer. Nonetheless, some aspects have to be pointed out.
One of the most critical terminological aspect that has to be addressed is represented by the definition of “dysphonia diagnosis”. “Dysphonia” is a symptom and a perceptual/acoustic sign, it is way too vague to be referred to as diagnosis. The term “diagnosis” refers to a pathological condition underlying dysphonia. The authors should consider this aspect and reformulate this definition along the text. I’d rather use “dysphonia-related diagnosis”
Detailed comments are listed below.
Abstract:
Line 29-31: this sentence is not clear and should be reformulated. I’d rather suggest something like: “Looking for medical assistance before a diagnosis of glottic cancer is associated with a better overall survival, while a diagnostic delay of more than 30 days from the first medical examination for dysphonia is associated with a worse outcome among in patients with laryngeal cancer”.
Introduction:
Line 46-48: “Advanced stage laryngeal cancer had a poor prognosis, with 5-year overall survival (OS) rate of 43% while more than 80% survive longer than 5 years in early stage cases “: further epidemiological references are required.
Line 57: “dysphonia diagnosis” please reformulate.
Methods
Line 76-79: the issue of “patients with prior diagnosis of dysphonia” has to be addressed and reformulated. If patients that developed laryngeal cancer priorly looked for medical assistance for dysphonia, is it reasonably presumable that they had pre-cancerous lesions. Talking about “dysphonia diagnosis” is extremely vague and misleading. This aspect should be clarified.
Line 80-82: how was the 30 days cut-off identified? Was it literature based? Please clarify.
Line 91: Statistical Analysis
Did you test the Gaussian distribution of the continuous variables before applying t-tests? That aspect should be clarified.
Line 94-96: a little more detailed explanation about the meaning and the rationale of applying a propensity score would be desirable, readers will surely appreciate.
Results:
Line 132-134: you talk about “OS” and then “risk of mortality”. This sentence has to be reformulated and clarified. If you are referring to your primary outcome (OS rates) please make it clearer.
Line 135: Table 3. Please clarify what HR refer to. 5 years OS rates?
Discussion:
In general, English translation of the discussion section is quite poor and should be revised.
163: please use “suggests”, not “proved”.
164: “dysphonia diagnosis”: please reformulate.
165-166: the sentence “In glottic cancer patients with dysphonia before the diagnosis showed better OS in our study” has a wrong syntaxis and has to be reformulated
166: delete “initially”
171: happen ïƒ happens
176: dysphonia diagnosis: please reformulate
179-180: do you refer to your results? If yes, you should report such data in the results section as well. If not, the sentence is not so clear and has to be reformulated.
232-237: all these aspect have to be explained (or at least anticipated) in the methods section as well, in order to give the reader a clearer explanation of the “dysphonia diagnosis” issue.
Conclusions:
Line 246-247: “the diagnosis time longer than 30 days with dysphonia had a worse outcome among laryngeal cancer” please reformulate.
No further comments
Round 2
Reviewer 2 Report
to be accepted in the current edition
This manuscript is a resubmission of an earlier submission. The following is a list of the peer review reports and author responses from that submission.
Round 1
Reviewer 1 Report
Thank you for your submission entitled 'Seeking Medical Assistance for Dysphonia Improves Survival Rate of Laryngeal Cancer: Real-world Evidence'
Minor revisions
Given the nature of the study, only association versus causation can be implied. Thus, the title should be changed to:
'Seeking Medical Assistance for Dysphonia is Associated with an Improved Survival Rate in Laryngeal Cancer: Real-world Evidence'
Reviewer 2 Report
In this paper, authors stated that finding dysphonia earlier improve survival rate of laryngeal cancer. I think that this article is too limited and too preliminary. Therefore, I cannot recommend this paper for publication in Cancers.
My comments in details are the following:
- In this article title, why can authors state ‘Real-world Evidence’? The results in this study have been limited to Taiwan. Title is overstated.
- Many patients who are diagnosed with laryngeal cancer are smokers, which is the most knowing as the risk factor. However, authors have used database without information about smoking habit. Authors should present the results including information with smoking details in this study. Moreover, author should state about smoking risk factor in introduction section properly.
- In Materials and Methods, there are including no need sentences from the first to third paragraph. All authors should read this article more carefully.
Reviewer 3 Report
The manuscript is very interesting and exciting. I understand that PD+ group did have a significantly better 5-year survival comparing with PD- group. Thus, authors concluded that seeking medical assistance for dysphonia before glottic cancer been diagnosed did improve the survival rate of glottic cancer. It is a certain. But, why?
I think that authors should clarify the content of medical assistance of dysphonia. Does the medical assistance involve antismoking education, etc?
Reviewer 4 Report
The authors analyze the OS of a larger group of throat cancer patients selected from a national database for the presence/absence of dysphonia. The quality and completeness of the collected data is questionable, which the authors confirm in the description of the study limitations. The use of ICD-9 CM codes is nowadays unusual (in the study, some codes are incorrectly listed). There are many methodological shortcomings, including statistical, that affect the relevance of the presented results. The discussion should be more focused. Cited references are often not appropriately chosen for the purpose. The text is written awkwardly and English requires a thorough examination and significant corrections. Abbreviations (e.g. OS) are not consistently used throughout the text.
Specific comments:
Simple Summary, line 4: Use abbreviation for overall survival that was introduced few lines above!
Abstract/Methods: Indicate first which groups of patients will be compared (than only followed a statement on a propensity score matching).
Introduction:
- Replace ref 1 with more recent one!
- The study results from reference 2 are not the most appropriate for the intended argumentation: in this study patients are divided into 2 groups according to whether they have localized cancer or regional disease – advanced T-stage localized cancer without regional metastasis also fall into the "advanced tumor" category.
Materials:
- Delete paragraphs 1-3! (i.e. journal’s instructions how to write the Matherials and Methods section).
- The authors appear to have used for patients selection a sample database (the Longitudinal Health Insurance Database) which is part of the larger, nationwide NHIRD database. They do not provide relevant data on the representativeness of this smaller database.
- What exactly does “first primary laryngeal cancer” mean? The first of several consecutive primary laryngeal or the first of several primary cancers in a patient or something else?
- ICD-9 CM codes 160.1-160.9… indicate malignancies of the nasal cavity and paranasal sinuses; you probably meant codes 161.1-161.9, which designate laryngeal malignancies!
- Listed codes (202 – 784) implies that patients’ visits (before diagnosing laryngeal cancer) were due to variety of larynx related disease conditions and not only due to hoarseness as you wrote in the text! (if the latter were the case, all patients included in the study would have some degree of dysphonia and there would be no PD- group in your study).
- The DDI interval should also be analyzed as a continuous variable.
- References 21-23 should be replaced by more appropriate ones.
- Is the information on the TNM stage and treatment intent available? (or info coded as early vs. advanced non-metastatic tumors; or as localized vs. regional tumors; curative vs. palliative/no treatment) – use these data in analyses if available!
Results:
- First state the demographic characteristics of the selected patients, only then the division into PD+/PD- group and according to the length of DDI.
- How many patients had glottic and supraglottic (or other larynx sites) cancer?
- In table 3 information is missing for supraglottic (or other larynx sites) cancer group.
- The results of propensity score matching are should also be presented to confirm the comparability of the PD+/PD-!
- Did patients differ between the IM and PL groups in demographic and other characteristics?
- When limiting analysis to patients with glottis tumors (to study the difference between PD+ and PD- groups) or to PD+ patients (to study the impact of DDI), the multivariate analysis should be used to identify independent prognosticators for OS.
Discussion:
- Reference 5 does not deal with the incidence of laryngeal cancer!
- What’s the difference between “clinical tumor characteristics and nodular stage”?
- Several studies confirmed (and several negate) correlations between delayed treatment and survival and the insurance status with survival. (see, e.g. ref 11; Žumer B et al. Radiat Oncol 2020;15:202; Naghavi AO et al. Cancer 2016;122:3529) The statement in the paragraph 1 (119-120) is in direct contrast to what is written in paragraph 5 (lines 164-173).
- Paragraph 2: delete last sentence or put it at the beginning of the paragraph.
- What would be “vocal demand”? And “medical factors” – do you mean treatment details?
Conclusions:
- Delete sentence 2: there is no mention of costs in the manuscript; I would suggest to include this topic in the discussion.